Exploring the effect of the triglyceride-glucose index on bone metabolism in prepubertal children, a retrospective study: insights from traditional methods and machine-learning-based bone remodeling prediction

Cao Shunshun 1
Chen Aolei 2
Song Botian 2
http://orcid.org/0000-0001-6710-7425 Hu Yangyang 2 209204@wzhealth.com
1 Pediatric Endocrinology, Genetics and Metabolism, The Second Affiliated Hospital of Wenzhou Medical University , Wenzhou , China
2 Reproductive Medicine Center, Obstetrics and Gynecology, The Second Affiliated Hospital of Wenzhou Medical University , Wenzhou , China
Menini Stefano
Electronic publication date: 2025 May 20
Publication date: 2025
Volume: 13
Electronic Location ID: e19483
Received 2025 Jan 21; Accepted 2025 Apr 26
Copyright: © 2025 Cao et al.
Copyright year: 2025
Copyright holder: Cao et al.
License: This is an open access article distributed under the terms of the Creative Commons Attribution License, which permits unrestricted use, distribution, reproduction and adaptation in any medium and for any purpose provided that it is properly attributed. For attribution, the original author(s), title, publication source (PeerJ) and either DOI or URL of the article must be cited.
License URL: https://creativecommons.org/licenses/by/4.0/

Keywords: Triglyceride glucose index, Bone turnover markers, Machine learning, Bone metabolism, Prepubertal children

Funding: Wenzhou Basic Scientific Research Project of China Y2023304 This work was supported by the Wenzhou Basic Scientific Research Project of China (No. Y2023304). The funders had no role in study design, data collection and analysis, decision to publish, or preparation of the manuscript.

==============================
Background

Childhood obesity poses a significant risk to bone health, but the impact of insulin resistance (IR) on bone metabolism in prepubertal children, as assessed by the triglyceride-glucose (TyG) index, remains underexplored. Bone turnover markers (BTMs) provide a non-invasive method for evaluating bone remodeling, but their relationship to obesity-related metabolic changes requires further study.

Methods

In this retrospective study of 332 prepubertal children (163 boys and 169 girls), we used multivariate linear regression and five machine learning (ML) algorithms to explore the association between the TyG index and BTMs, including β-C-terminal telopeptide of type 1 collagen (β-CTx), total procollagen type 1 N-terminal propeptide (T-P1NP), and N-terminal mid-fragment of osteocalcin (N-MID). The categorical boosting (CatBoost) models selected based on optimal performance metrics were interpreted using SHapley Additive exPlanation (SHAP) analysis to identify key features affecting prediction.

Results

The TyG index was negatively correlated with β-CTx, T-P1NP, and N-MID levels (P < 0.05), with a dose-response effect. The CatBoost model showed higher predictive accuracy and robustness, with the area under the receiver operating characteristic curve (AUROC) values of 0.782 (95% CI [0.68–0.885]), 0.789 (95% CI [0.691–0.874]), and 0.727 (95% CI [0.619–0.827]) for β-CTx, T-P1NP, and N-MID predictions, respectively. The SHAP analysis highlighted body mass index (BMI) and HbA1c as the key predictors.

Conclusions

The TyG index is a reliable predictor of bone metabolic disorders in prepubertal obese children, and the interpretable CatBoost model provides a cost-effective tool for early intervention. This study has important implications for prevention strategies for disorders of bone metabolism in prepubertal obese children to reduce the risk of skeletal fragility in adulthood or old age.

Introduction

The prevalence of childhood obesity has increased dramatically in all regions of the world over the past 2–3 years, with as many as one-third of children in Europe and the United States being overweight and obese (Chung, Krenek & Magge, 2023). Globally, obesity incurs an annual cost of nearly $2 trillion, with direct and indirect losses projected to reach $4 trillion by 2035, representing approximately 3% of the global gross domestic product (Dicken & Batterham, 2024). The TyG index assesses the body’s insulin sensitivity by combining triglyceride and fasting blood glucose levels (Dang et al., 2024). The triglyceride-glucose (TyG) index is a novel biomarker of insulin resistance (IR), an important risk factor for various metabolic diseases (Sun et al., 2025). Aslan Çin et al. (2020) demonstrated that the TyG index is a superior biomarker for assessing IR and the risk of metabolic syndrome compared to the homeostasis model assessment of IR (HOMA-IR). Children’s body size, puberty, nutrition, and metabolism are different from those of adults and reflect unique states, making it important to explore the relationship between the TyG index, obesity, and disease in children (Abdollahian et al., 2023; Badakhshan et al., 2023). Childhood obesity can lead to serious consequences affecting various organs and systems, such as the early onset of puberty, non-alcoholic fatty liver disease, diabetes mellitus (DM), and IR (Quarta et al., 2023; Aniśko, Siatkowski & Wójcik, 2024; Sohouli et al., 2024). However, its impact on the skeletal system remains particularly controversial (López-Peralta et al., 2022).

Bilinski, Paradowski & Sypniewska (2020) reviewed bone health and hyperglycemia in the pediatric population, stating that bone health in children can be assessed in different ways such as bone mineral density (BMD), bone strength, bone microarchitecture, and bone turnover, and that impaired regulation of glucose metabolism in children and adolescents, and IR can adversely affect bone health. Dole et al. (2024) demonstrated that obesity increases the risk of skeletal fragility, a vulnerability that stems from poor bone quality and may be caused by defects in the material properties of the bone matrix and that TGF-β signaling plays a key role in mediating the effects of obesity on the bone. Fuglsang-Nielsen et al. (2022) showed that abdominal obesity and type 2 DM (T2DM) decreased bone turnover, increased BMD, and increased fracture risk in participants. Lappe et al. (2015) concluded that a 10% increase in peak bone mass during childhood is associated with a 50% reduction in the risk of osteoporotic fractures in advanced age. Furthermore, they found that for each 1 standard deviation decrease in peak bone mass in childhood, the relative risk of fractures in later life increases by 2.6-fold. Bone mass accumulates rapidly during childhood and adolescence, reaching its peak around the age of twenty. Metabolic bone diseases, abnormal weight fluctuations, and the effects of certain medications can disrupt this accumulation process, potentially increasing the risk of osteoporosis and bone fragility in adulthood (Thom et al., 2023). Consequently, osteoporosis prevention efforts should ideally begin in childhood. Tahani et al. (2021) demonstrated that obese children exhibit higher bone mineral content (BMC) and BMD compared to their normal-weight peers. Liang et al. (2022) demonstrated that obese children have a higher risk of supracondylar humeral fractures compared to children of healthy weight and found a correlation between obesity, BMD, and fracture risk.

Rinonapoli et al. (2021) reviewed evidence indicating that obesity is associated with an increase in BMD, likely due to mechanical loading effects and elevated estrogen levels. However, despite the increase in BMD, individuals with obesity have a higher risk of fractures, suggesting that BMD values alone are insufficient to accurately assess fracture risk. Dual-energy X-ray absorptiometry (DXA) and quantitative computed tomography (QCT) are commonly used methods for measuring BMD and BMC. However, their clinical application for long-term monitoring of bone health in children is limited due to potential radiation risks and the high cost of these examinations (Zhang et al., 2023). Bone turnover markers (BTMs) are biochemical indicators of bone metabolism that can be detected earlier in the bloodstream than changes in BMD, reflecting the bone remodeling rate and metabolism state (Vasikaran et al., 2024). However, clinical data remain insufficient regarding the use of BTMs to monitor the effects of childhood obesity on bone metabolism and bone remodeling.

The current studies have focused on the association between TyG index and stroke and cardiovascular disease in adults, whereas the relationship between TyG index and bone remodeling and metabolism in prepubertal children is unclear (Chen et al., 2024). Therefore, the purpose of our study was twofold: (1) to investigate the association between TyG index and BTMs (β-C-terminal telopeptide of type 1 collagen (β-CTx), total procollagen type 1 N-terminal propeptide (T-P1NP), and N-terminal mid-fragment of osteocalcin (N-MID)); and (2) the TyG index was combined with other variables to construct an interpretable machine learning (ML) model to predict bone metabolism in prepubertal children for monitoring bone health.

Materials and Methods

Data selection and study design

A total of 356 prepubertal children who visited the Department of Pediatric Endocrinology at the Second Affiliated Hospital of Wenzhou Medical University between June 2021 and October 2023 were retrospectively analyzed. Of these, 332 participants met the inclusion criteria, comprising 163 boys and 169 girls, aged 7–11 years. To avoid the influence of puberty on the results of the study, we chose children in the Tanner stage 1 as the study participants (Eckert-Lind et al., 2020). This is because the rapid skeletal development of children during puberty is accompanied by an increased secretion of growth hormones and sex hormones (e.g., testosterone and estrogen), which play key roles in bone formation, remodeling, and metabolism. Written informed consent was obtained from both participants and their guardians. This study adhered to the principles of the Declaration of Helsinki and was approved by the Medical Ethics Committee of the Second Affiliated Hospital of Wenzhou Medical University (Approval No. 2024-K-031-02).

Inclusion criteria were as follows: (1) written informed consent was obtained from both participants and guardians, (2) participants cooperated with medical history taking, anthropometric measurements, and fasting blood sample collection, and (3) Tanner stage 1 children aged 7–11 years. Exclusion criteria included: (1) missing key data, (2) presence of metabolic bone disease, (3) chromosomal or genetic abnormalities affecting endocrine or bone metabolism, (4) history of bone fracture within the past year, (5) use of medications influencing bone metabolism, (6) Tanner stages 2–5, and (7) chronic systemic diseases.

Grouping and variable definitions

According to the standardized growth charts from the Centers for Disease Control and Prevention, adjusted for age and gender, we defined a healthy weight as a body mass index (BMI) between the 5th and 84th percentiles, overweight as a BMI between the 85th and 94th percentiles, and obesity as a BMI at or above the 95th percentile, as described by Peinado Fabregat, Saynina & Sanders (2023). We categorized the participants into three groups: 135 in the healthy weight group (64 boys and 71 girls), 86 in the overweight group (41 boys and 45 girls), and 111 in the obese group (58 boys and 53 girls).

Since there was no clear cut-off value for BTMs, we classified β-CTx (threshold: 488.345 pg/ml), T-P1NP (threshold: 374.000 ng/ml), and N-MID (threshold: 29.280 ng/ml) according to the median level of BTMs in the participants to suggest enhanced or inhibited bone metabolism and remodeling, respectively. The TyG index is calculated using the formula: ln[fasting triglycerides (mg/dL) × fasting glucose (mg/dL)/2] (Tao et al., 2022). Similar to the TyG index, the TyG-BMI index is a novel biomarker for IR, calculated by multiplying the TyG index by BMI (Huang et al., 2023). Tang et al. (2022) demonstrated that systemic immune inflammation (SII) is a reliable and stable hematological indicator of the body’s systemic immune and inflammatory status, calculated using the formula: platelet count × neutrophil count/lymphocyte count. The HOMA-IR was calculated using the formula: HOMA-IR = Fasting Insulin (μU/mL) × Fasting Glucose (mmol/L)/22.5 (Tahapary et al., 2022). Spexin, a recently identified adipokine belonging to the galanin/kisspeptin/spexin peptide family, plays a significant role in the pathophysiology of obesity-induced IR and T2DM (Fang et al., 2022). Fibroblast growth factor 23 (FGF23) is an osteogenic hormone produced and secreted by the skeletal system. Alongside parathyroid hormone (PTH), 1,25-dihydroxyvitamin D (1,25(OH)2D), and calcitonin, FGF23 plays a critical role in regulating phosphate and calcium homeostasis across the bone, kidneys, and gastrointestinal tract to support optimal bone mineralization (Cipriani et al., 2022).

Missing value handles

In medical research, there are often missing values in medical data, which can result in wasted sample size if only complete variables are considered for analysis, and statistical analyses that ignore missing values have the potential to introduce bias in parameter estimation (Cao & Hu, 2024). Imputation of missing values plays a vital role in maintaining the integrity of the data and improving the validity of the study results. In this study, we used the R package “mice” to multiply impute variables with less than 15% missing data (Blazek et al., 2021). Variables with more than 15% missing data were excluded from the analysis.

Feature engineering and modeling strategies

Based on previous studies (Schini et al., 2023; Vasikaran et al., 2024) and expert opinions from pediatric and orthopedic specialists at the Second Affiliated Hospital of Wenzhou Medical University, we selected demographics, laboratory parameters, and anthropometric measurements as the initial model features for this study, comprising a total of 26 variables (Data S1). We screened the model variables through correlation analysis and near-zero variance tests to mitigate multicollinearity and overfitting. Based on the results of the correlation analysis (correlation coefficient threshold > 0.85) and the comparison of the explanatory power of the variables on the target variables, we removed four redundant variables: waist circumference, TyG-BMI index, insulin, and triglycerides, and ended up with a dataset containing 22 features and three target variables (β-CTx, T-P1NP, and N-MID). The dataset includes one categorical feature and 21 continuous features. To more robustly evaluate the performance of the model in unknown data, we randomly split the original predictive BTMs dataset into a training set and a test set based on 80:20. We used a five-fold cross-validation and grid search to determine the optimal combination of hyperparameters for the model, trained the model using the training set data, and evaluated the model’s performance in predicting BTMs in the test set data (Chen et al., 2023). The data were normalized using Scikit-Learn’s StandardScaler, and categorical variables were transformed into numerical form through one-hot encoding (Singh et al., 2020, 2023). Different ML algorithms have different applicability due to their ability to handle category variables, data nonlinearities, and computational efficacy. We employed five ML algorithms—support vector machine (SVM), Random Forest (RF), extreme gradient boosting (XGBoost), light gradient boosting machine (LGBM), and categorical boosting (CatBoost)—to investigate the relationship between TyG indexes and BTMs, including β-CTx, T-P1NP, and N-MID. After evaluating seven performance metrics for each ML model, the model that best captured the relationship between the TyG index and BTMs was selected and interpreted using SHapley Additive exPlanation (SHAP) analysis.

SHAP-based model interpretability approach

With the extensive application of ML and artificial intelligence in healthcare, model interpretability has emerged as a critical topic in both research and practical applications. The high complexity of these models, along with the nonlinearity of their features, often complicates the understanding of their internal decision logic and influencing factors. SHAP-based interpretation methods offer a novel approach for visualizing and quantifying model behavior, enhancing the transparency and usability of complex predictive models in healthcare settings (Fan et al., 2023). The SHAP value quantifies the importance of each feature by calculating its marginal contribution across various feature combinations, ensuring fair consideration of each feature’s role in different permutations (Wang et al., 2021). This approach enhances our understanding of how individual features influence model decisions. In the healthcare domain, SHAP values facilitate insights into how specific features impact a given patient’s diagnostic outcome, thereby supporting personalized diagnosis and treatment.

Statistical analysis

The study sample size was determined using the pmsampsize R package (Riley et al., 2020). Based on data distribution, normally distributed data were represented as mean ± standard deviation (SD), while non-normally distributed data were expressed as median and interquartile range (25%, 75%). The Kruskal-Wallis H test was used for comparisons across three groups, with categorical variables presented as frequencies (percentages) and analyzed using the Pearson chi-square test for group comparisons. Multivariate linear regression assessed the correlation between the TyG index and BTMs. The performance of the ML model was evaluated using metrics including the area under the receiver operating characteristic curve (AUROC), accuracy, precision, recall, F1 score, brier score, and the area under the P–R curve (AP). Statistical significance was defined as a two-sided P-value < 0.05. Data analysis was performed using R software version 4.3.1, with the following packages: tidyverse, dplyr, tsummary, ggplot2, tableone, glmnet, car, MASS, psych, rms, ggExtra, mice, pmsampsize, and ggpubr. ML analyses were conducted in Python version 3.11.5, utilizing the scikit-learn library (version 1.2.2).

Results

Baseline characteristics of the study population

Figure 1 shows the process for inclusion and exclusion of participants, as well as the flow of the study design, which resulted in the inclusion of 332 prepubertal children as the study sample. Table 1 summarizes the baseline characteristics of participants grouped according to BMI. Gender, age, and 1,25(OH)2D levels were not statistically different when compared between the healthy weight, overweight, and obese groups, respectively (P > 0.05). In the obese group, height, high-density lipoprotein (HDL), Spexin, β-CTx, T-P1NP, and N-MID levels were lower than in the healthy weight and overweight groups, while the rest of the variables were higher than in the healthy weight and overweight groups, and the difference was statistically significant (P < 0.05).

Figure 1 Flowchart of participant inclusion, exclusion criteria screening, and study design.

Table 1 Baseline characteristics of participants grouped by BMI.

Characteristics	Healthy weight group
(N = 135)	Overweight group
(N = 86)	Obese group
(N = 111)	P value	
Gender (n)				0.700	
Boys	64 (47.41%)	41 (47.67%)	58 (52.25%)		
Girls	71 (52.59%)	45 (52.33%)	53 (47.75%)		
Age (years)	9.00 (8.00, 10.00)	9.00 (8.00, 10.00)	9.00 (8.00, 10.00)	0.300	
Weight (kg)	35.40 (32.10, 39.55)	41.20 (35.98, 47.28)	46.30 (40.65, 52.55)	<0.001	
Height (cm)	145.50 (141.00, 149.05)	142.65 (136.33, 150.08)	140.30 (135.00, 146.75)	<0.001	
BMI (kg/m2)	16.69 (15.67, 18.07)	20.25 (19.25, 21.23)	23.26 (21.56, 25.46)	<0.001	
WC (cm)	61.70 (58.80, 66.00)	69.15 (66.40, 74.28)	78.40 (72.75, 83.10)	<0.001	
Total BMD (g/cm2)	0.82 (0.78, 0.87)	0.83 (0.78, 0.92)	0.86 (0.82, 0.90)	<0.001	
TBLH BMD (g/cm2)	0.71 (0.66, 0.75)	0.73 (0.68, 0.76)	0.81 (0.76, 0.87)	<0.001	
TBLH BMD Z -score	1.32 (0.95, 1.69)	1.61 (1.23, 1.90)	2.00 (1.62, 2.37)	<0.001	
Lumbar spine BMD (g/cm2)	0.71 (0.66, 0.76)	0.74 (0.68, 0.79)	0.79 (0.71, 0.86)	<0.001	
Total fat percentage (%)	26.70 (23.05, 30.10)	33.90 (30.33, 37.40)	38.40 (35.10, 41.75)	<0.001	
SII	321.43 (226.44, 391.19)	383.60 (290.79, 487.39)	403.13 (329.94, 512.09)	<0.001	
HS- CRP (mg/L)	0.91 (0.56, 1.15)	1.33 (1.00, 1.69)	1.95 (1.55, 2.37)	<0.001	
HDL (mmol/L)	1.69 (1.35, 2.05)	1.18 (1.01, 1.39)	0.78 (0.63, 1.03)	<0.001	
LDL (mmol/L)	1.58 (1.08, 2.08)	2.52 (2.04, 2.97)	3.11 (2.62, 3.91)	<0.001	
TC (mmol/L)	3.25 (2.68, 3.84)	3.95 (3.40, 4.32)	4.59 (4.08, 5.03)	<0.001	
FBG (mmol/L)	4.82 (4.53, 5.10)	5.45 (5.11, 5.57)	5.56 (5.50, 5.74)	<0.001	
TyG index	8.36 (8.18, 8.51)	8.82 (8.67, 8.94)	9.29 (9.10, 9.49)	<0.001	
HOMA-IR	2.26 (1.87, 2.61)	3.70 (3.09, 4.18)	5.75 (5.01, 6.47)	<0.001	
HbA1c (%)	4.61 (4.32, 5.01)	5.21 (4.95, 5.38)	5.59 (5.13, 6.05)	<0.001	
Spexin (ng/ml)	0.55 (0.46, 0.63)	0.47 (0.37, 0.56)	0.34 (0.25, 0.43)	<0.001	
FGF23 (ng/ml)	48.50 (43.41, 52.90)	47.53 (41.35, 54.77)	51.60 (41.88, 63.94)	0.017	
β- CTx (pg/ml)	524.19 (482.02, 566.15)	488.91 (449.63, 543.20)	429.37 (379.67, 487.86)	<0.001	
T-P1NP (ng/ml)	410.73 (340.09, 504.30)	402.17 (297.74, 468.59)	312.24 (233.62, 398.21)	<0.001	
N-MID (ng/ml)	32.73 (26.52, 40.63)	28.47 (22.76, 32.91)	25.68 (19.72, 32.52)	<0.001	
1,25(OH)2D (ng/ml)	61.00 (53.18, 71.63)	59.27 (52.98, 66.83)	58.44 (51.70, 66.07)	0.300	
Note:

BMI, body mass index; WC, waist circumference; TBLH, total body less head; BMD, bone mineral density; SII, systemic immune inflammation; HS- CRP, hypersensitive C-reactive protein; HDL, high-density lipoprotein; LDL, low-density lipoprotein; TC, total cholesterol; FBG, fasting blood glucose; TyG, triglyceride glucose; HOMA, homeostasis model assessment; FGF23, fibroblast growth factor 23; β-CTx, β-C-terminal telopeptide of type 1 collagen; T-P1NP, total procollagen type 1 N-terminal propeptide; N-MID, N-terminal mid-fragment of osteocalcin; 1,25(OH)2D, 1,25-dihydroxyvitamin D.

Dose-response effects of TyG index and BTMs

Table 2 summarizes the results of the multivariate linear regression analysis of the relationship between the TyG index and BTMs. Initially, the unadjusted model showed that the TyG index was associated with the response variables β-CTx (β = −75.98, 95% CI [−90.85 to −61.10], P < 0.001), T-P1NP (β = −91.99, 95% CI [−115.24 to −68.73], P < 0.001), and N-MID (β = −6.62, 95% CI [−8.54 to −4.70], P < 0.001), respectively, and was negatively correlated. In model 2, this association remained statistically significant even after adjusting for gender and age factors, β-CTx (β = −76.00, 95% CI [−90.90 to −61.11], P < 0.001), T-P1NP (β = −92.03, 95% CI [−115.23 to −68.83], P < 0.001), N-MID (β = −6.61, 95% CI [−8.52 to −4.70], P < 0.001). In Model 3, after adjusting for more covariates, for each unit increase in the TyG index, β-CTx decreased by 54.66 pg/ml (β = −54.66, 95% CI [−75.91 to −33.42], P < 0.001), T-P1NP decreased by 72.38 ng/ml (β = −72.38, 95% CI [−105.60 to −39.17], P < 0.001), and N-MID decreased by 4.21 ng/ml (β = −4.21, 95% CI [−6.97 to −1.46], P = 0.003), and the correlation remained significant. In Model 3, we observed that individuals in the highest quartile of the TyG index exhibited reductions in β-CTx, T-P1NP, and N-MID levels compared to those in the lowest quartile of the TyG index. Specifically, β-CTx decreased by 62.18 pg/mL (β = −62.18, 95% CI [−89.22 to −35.14], P < 0.001), T-P1NP decreased by 90.35 ng/mL (β = −90.35, 95% CI [−132.47 to −48.23], P < 0.001), and N-MID decreased by 5.07 ng/mL (β = −5.07, 95% CI [−8.56 to −1.58], P = 0.005). In model 3, there was a significant dose-response effect of the TyG index with β-CTx, T-P1NP, and N-MID when the TyG index was ≥8.74, ≥8.44, and ≥8.74, respectively (P < 0.05). Normality, independence, homoscedasticity, and linearity assumptions of the multivariate linear regression equations were evaluated using Q-Q plots, the Durbin-Watson test, the non-constant variance score test, and the variance inflation factor (VIF), respectively. Results indicate that all model assumptions are met, supporting the model’s robustness.

Table 2 Multivariate linear regression analysis of the relationship between the TyG index and serum BTMs.

Exposure	Model 1	Model 2	Model 3	
β (95%CI) P value	β (95%CI) P value	β (95%CI) P value	
Response variable: β-CTx				
TyG index (continuous)	−75.98 [−90.85 to −61.10] <0.001	−76.00 [−90.90 to −61.11] <0.001	−54.66 [−75.91 to −33.42] <0.001	
TyG index (quartile)				
Q1 (≥7.79, <8.44)	Reference	Reference	Reference	
Q2 (≥8.44, <8.74)	−12.20 [−31.46 to 7.06] 0.215	−11.43 [−30.77 to 7.90] 0.247	−8.22 [−27.76 to 11.32] 0.410	
Q3 (≥8.74, <9.10)	−54.97 [−74.12 to −35.83] <0.001	−55.06 [−74.23 to −35.89] <0.001	−38.48 [−62.04 to −14.91] 0.002	
Q4 (≥9.10, <9.66)	−86.79 [−106.11 to −67.46] <0.001	−88.77 [−106.10 to −67.44] <0.001	−62.18 [−89.22 to −35.14] <0.001	
P for trend	<0.001	<0.001	<0.001	
Response variable: T-P1NP				
TyG index (continuous)	−91.99 [−115.24 to −68.73] <0.001	−92.03 [−115.23 to −68.83] <0.001	−72.38 [−105.60 to −39.17] <0.001	
TyG index (quartile)				
Q1 (≥7.79, <8.44)	Reference	Reference	Reference	
Q2 (≥8.44, <8.74)	−36.93 [−67.03 to −6.84] 0.017	−34.79 [−64.90 to −4.68] 0.024	−33.38 [−63.82 to −2.93] 0.032	
Q3 (≥8.74, <9.10)	−63.80 [−93.71 to −33.88] <0.001	−62.81 [−92.66 to −32.96] <0.001	−47.59 [−84.31 to −10.88] 0.012	
Q4 (≥9.10, <9.66)	−114.43 [−144.62 to −84.23] <0.001	−114.58 [−144.68 to −84.47] <0.001	−90.35 [−132.47 to −48.23] <0.001	
P for trend	<0.001	<0.001	<0.001	
Response variable: N-MID				
TyG index (continuous)	−6.62 [−8.54 to −4.70] <0.001	−6.61 [−8.52 to −4.70] <0.001	−4.21 [−6.97 to −1.46] 0.003	
TyG index (quartile)				
Q1 (≥7.79, <8.44)	Reference	Reference	Reference	
Q2 (≥8.44, <8.74)	−1.73 [−4.21 to 0.75] 0.172	−1.88 [−4.36 to 0.59] 0.137	−1.36 [−3.88 to 1.16] 0.291	
Q3 (≥8.74, <9.10)	−5.89 [−8.35 to −3.43] <0.001	−5.88 [−8.33 to −3.42] <0.001	−3.95 [−6.99 to −0.91] 0.011	
Q4 (≥9.10, <9.66)	−7.66 [−10.15 to −5.18] <0.001	−7.66 [−10.14 to −5.19] <0.001	−5.07 [−8.56 to −1.58] 0.005	
P for trend	<0.001	<0.001	0.003	
Note:

CI, confidence intervals; Model 1, no covariate adjustment; Model 2, Adjusted for gender and age; Model 3, adjusted for gender, age, weight, SII, and LDL.

Comparison of ML model performance metrics

Table 3 summarizes fifteen ML models’ performance metrics for predicting BTMs. We trained fifteen ML models predicting β-CTx, T-P1NP, and N-MID based on the training set data and tested each model on the test set data for performance evaluation metrics including AUROC, accuracy, precision, recall, F1-score, Brier score, and AP, and 1,000 times resampling using Bootstrap to improve the robustness of the model predictions. Based on the confusion matrix and ROC curves, the CatBoost model has higher accuracy and robustness in predicting β-CTx, T-P1NP, and N-MID compared to other models (Fig. 2, Figs. S1–S4). The AUROC (95% CI) for predicting β- CTx, T-P1NP and N-MID were 0.782 [0.683–0.885], 0.789 [0.691–0.874], and 0.727 [0.619–0.827], respectively.

Table 3 Evaluation of performance metrics of five ML models for predicting BTMs (β-CTx/T-P1NP/N-MID).

Characteristics	SVM	RF	XGBoost	LGBM	CatBoost	
AUROC	0.716/0.683/0.668	0.716/0.764/0.695	0.721/0.688/0.711	0.707/0.675/0.715	0.782/0.789/0.727	
95% CI	[0.612–0.822]/
[0.576–0.792]/
[0.550–0.780]	[0.610–0.817]/
[0.668–0.858]/
[0.585–0.796]	[0.618–0.820]/
[0.574–0.798]/
[0.607–0.814]	[0.600–0.803]/
[0.558–0.782]/
[0.604–0.817]	[0.683–0.885]/
[0.691–0.874]/
[0.619–0.827]	
Accuracy (%)	85.07/94.03/82.09	88.06/88.06/92.54	85.07/91.04/94.03	89.55/89.55/89.55	92.54/92.54/94.03	
Precision (%)	67.74/90.00/63.64	75.00/80.95/86.49	76.19/81.82/84.00	81.08/77.42/72.00	83.87/88.37/90.24	
Recall (%)	100/100/100	100/100/100	100/100/100	100/100/100	100/100/100	
F1 score	0.808/0.947/0.778	0.857/0.895/0.928	0.865/0.900/0.913	0.896/0.873/0.837	0.912/0.938/0.949	
Brier score	0.218/0.232/0.234	0.215/0.200/0.223	0.218/0.224/0.232	0.217/0.227/0.233	0.220/0.228/0.231	
AP	0.713/0.743/0.695	0.674/0.832/0.699	0.704/0.638/0.720	0.697/0.620/0.769	0.741/0.767/0.676	
Note:

SVM, support vector machine; RF, random forest; XGBoost, extreme gradient boosting; LGBM, light gradient boosting machine; CatBoost, category boosting; AUROC, area under the receiver operating characteristic curve; AP, area under the P–R curve.

Figure 2 The CatBoost model predicting BTMs (β-CTx/T-P1NP/N-MID).

(A, D, G) The confusion matrices of CatBoost models predicting β-CTx, T-P1NP, and N-MID, respectively. (B, E, H) The ROC curves of CatBoost models predicting β-CTx, T-P1NP, and N-MID, respectively. (C, F, I) The P-R curves of CatBoost models predicting β-CTx, T-P1NP, and N-MID, respectively.

SHAP features importance and modeling decisions

The SHAP global summary plot shows the positive and negative impact of all features in the CatBoost model on predicting β-CTx, T-P1NP, and N-MID, sorted by the importance of each feature’s contribution to the model output. The model SHAP values suggested a negative correlation between TyG index and BMI for the prediction of BTMs (β-CTx, T-P1NP, N-MID) (Figs. 3A, 4A, and 5A). The SHAP summary plot suggests that the most important features for predicting β-CTx, T-P1NP, and N-MID in the CatBoost model are BMI, BMI, and HbA1c, respectively (Figs. 3B, 4B, and 5B). The SHAP individual decision plot reveals the individual prediction of BTMs (β-CTx, T-P1NP, N-MID) for each feature in the CatBoost model (Figs. 3C, 4C, and 5C). In the decision plot, the gray vertical line represents the base value of the CatBoost model’s prediction of BTMs, the red curve indicates a positive prediction of BTMs, and the blue curve indicates a negative prediction of BTMs. The decision curves reflect the specific contribution of each feature to the CatBoost model’s BTMs prediction and label the SHAP value of each feature.

Figure 3 The CatBoost model predicts the SHAP interpretation of β-CTx (top 15 features).

(A) The SHAP global summary plot. (B) The SHAP features importance. (C) The SHAP individual decision plot. (D) The SHAP top 50 individual decision plot.

Figure 4 The CatBoost model predicts the SHAP interpretation of T-P1NP (top 15 features).

(A) The SHAP global summary plot. (B) The SHAP features importance. (C) The SHAP individual decision plot. (D) The SHAP top 50 individual decision plot.

Figure 5 The CatBoost model predicts the SHAP interpretation of N-MID (top 15 features).

(A) The SHAP global summary plot. (B) The SHAP features importance. (C) The SHAP individual decision plot. (D) The SHAP top 50 individual decision plot.

Discussion

Our study explored the relationship between the TyG index and bone metabolism and remodeling in prepubertal children, revealing some important findings: (1) The TyG index was negatively correlated with β-CTx, T-P1NP, and N-MID, and this relationship remained significant when adjusting for more confounding factors. This demonstrates that IR inhibits bone metabolism and bone remodeling in prepubertal children. (2) Significant dose-response effects of the TyG index with β-CTx, T-P1NP, and N-MID were observed when the TyG index was ≥8.74, ≥8.44, and ≥8.74, respectively. This suggests that there exists a threshold for IR to inhibit bone metabolism and bone remodeling in prepubertal children, with significant dose-response effects on bone remodeling and bone metabolism only when TyG index ≥8.74, ≥8.44, and ≥8.74, respectively. (3) The TyG index can be used as an important independent predictor for constructing the CatBoost model to predict the state of bone metabolism and bone remodeling in prepubertal children, which is important for clinical monitoring of bone health.

This is the first study on prepubertal children to employ an interpretable ML framework in a retrospective cohort study to investigate the association between the TyG index and BTMs (β-CTx, T-P1NP, and N-MID). The SHAP-based CatBoost model combines powerful predictive performance and interpretation capabilities for bone metabolism monitoring scenarios in prepubertal children that require high-accuracy prediction and model transparency. This combination enhances the usability and credibility of the model and provides strong support for clinical decision-making. In contrast to the increased BMD in children with high BMI found by Tahani et al. (2021), the present study suggests that IR may lead to negative regulation of bone metabolism from the perspective of bone metabolism markers, providing a new perspective to explain the high fracture risk in children with high BMI. Although similar reports of using ML to predict bone health in prepubertal children using BTMs have not been found, we still compared them with ML models that predict osteoporosis in adults. Baik et al. (2024) constructed an ML diagnostic model for predicting osteoporosis in adults using BTM and demographic variables, and their best model was LGBM with an internally validated AUC of 0.706, whereas the CatBoost model AUCs of 0.782, 0.789, and 0.727 suggested better predictive performance and robustness, respectively.

The CatBoost model as our optimal model for predicting BTMs may be related to its many advantages. The CatBoost model has a variety of regularization means (e.g., using Ordered Boosting) that can effectively prevent data overfitting, especially when the data size is small (Hu et al., 2024). For medical data, where the sample size for training and validating the model is often small, the generalization ability of the model is crucial for the generalization of the findings, and the CatBoost model also has the advantage of being efficient, robust, and dealing with categorical features. In addition, the CatBoost model provides a variety of model interpretability tools (e.g., SHAP) that can help users understand the basis of the model’s decisions, which is particularly important in the field of medical diagnosis, where clinicians need to understand the reasons for the model’s judgments (Fieggen et al., 2022).

Chen et al. (2024) suggested that based on the available evidence, the sensitivity and specificity of TyG for the diagnosis of IR were 96.5% and 85.0%, respectively, and that TyG could be used as a reliable and easily evaluated alternative measure of IR. Studies have shown that T2DM increases BMD yet also raises fracture risk, likely due to factors such as abnormal bone metabolism, reduced bone turnover, lower bone mass, and altered bone microarchitecture (Sun et al., 2023). These findings suggest that BMD alone is insufficient as an indicator for assessing osteoporosis risk. Gkastaris et al. (2020) demonstrated that the positive impact of increased mechanical loading on BMD associated with obesity may depend on the site of weight-bearing. Conversely, low-grade systemic inflammation and elevated bone marrow adipogenesis resulting from obesity may exert negative effects on bone health (Gkastaris et al., 2020). These findings align with our study, which demonstrated that BMI was positively correlated with the SII, lumbar BMD, and total body less head (TBLH) BMD Z-scores while being negatively correlated with serum levels of β-CTx, T-P1NP, and N-MID. Furthermore, the SHAP summary plot revealed that elevated TyG index and BMI values were negatively associated with the predictive values of β-CTx, T-P1NP, and N-MID. Rand et al. (2023) showed that baseline bone-specific alkaline phosphatase, β-CTx, osteocalcin, and T-P1NP levels in healthy children and adolescents were positively correlated with subsequent changes in total BMD/BMC. The foundation of skeletal fragility in old age is partially established during childhood and adolescence, and the accumulation of bone mass during this period contributes largely to peak bone mass. Combined with our findings, we hypothesize that childhood obesity may be associated with the risk of skeletal fragility in adulthood or old age.

1,25(OH)2D and phosphorus are essential components for maintaining various biological functions in humans, playing critical roles in energy metabolism and bone mineralization (Jin, Bertholf & Yi, 2023). Phosphorus deficiency can lead to musculoskeletal disorders, whereas phosphorus excess may cause ectopic calcification of tissues and organs, increasing mortality risk. The regulation of phosphate homeostasis is primarily controlled by PTH, 1,25(OH)2D, and fibroblast growth factor 23 (FGF23) (Peacock, 2021). Studies have demonstrated associations between FGF23, obesity, and kidney disease, with FGF23 positively correlated with total body fat percentage, visceral fat, and male-pattern adipose tissue while showing a negative correlation with lean body mass (Parente et al., 2023). Additionally, Karampatsou et al. (2022) concluded that FGF23, osteoblasts, and osteosclerotic proteins are influenced by overweight and obesity, varying with BMI, and highlighted the interplay between adipose and bone tissues. These findings align with our results, which revealed that FGF23 levels were higher in the obese group compared to the overweight and healthy weight groups.

Our study has several strengths and limitations. First, interpretable ML aims to strike a balance between prediction accuracy and model comprehensibility, making it a powerful tool for researchers and practitioners, particularly in contexts where the transparency and reliability of results are critical. Second, we constructed a total of fifteen models using five ML algorithms and ultimately selected the best-performing CatBoost model. This model was further combined with a traditional linear regression model to overcome the drawback of linear models that are difficult to capture nonlinear variables and provide a comprehensive analysis of the relationship between the TyG index and BTMs. Third, the CatBoost model we constructed can provide a reliable skeletal metabolism monitoring tool for prepubertal children with IR, which has important clinical applications. Although our model performs well, there may be some population limitations due to geographic constraints of the sample source. Future studies should introduce cross-regional and multi-ethnic samples to improve the model’s generalization ability.

Conclusions

This study demonstrated for the first time that the TyG index can be used as a potential biomarker for assessing bone metabolic status in prepubertal children and provided accurate and highly interpretable predictive results by CatBoost modeling. In the future, its applicability in multicenter samples should be further validated and clinical tools based on this model should be developed. Applying the methods of this study to the management of bone health in prepubertal obese children may significantly reduce the risk of skeletal fragility in adulthood or old age.

Supplemental Information

Supplemental Information 1 R code and raw data for Tables 1 and 2.

Supplemental Information 2 Part of the Python code and the raw data for Table 3 and Figures 2-9.

Supplemental Information 3 Part of the Python code.

Supplemental Information 4 Categorized Data R Code.

Supplemental Information 5 The SVM model predicting BTMs (β-CTx/ T-P1NP/ N-MID).

(A), (D), and (G) Denote the confusion matrices of SVM models predicting β-CTx, T-P1NP, and N-MID, respectively. (B), (E), and (H) Denote the ROC curves of SVM models predicting β-CTx, T-P1NP, and N-MID, respectively. (C), (F), and (I) Denote the P-R curves of SVM models predicting β-CTx, T-P1NP, and N-MID, respectively.

Supplemental Information 6 The XGBoost model predicting BTMs (β-CTx/ T-P1NP/ N-MID).

(A), (D), and (G) Denote the confusion matrices of XGBoost models predicting β-CTx, T-P1NP, and N-MID, respectively. (B), (E), and (H) Denote the ROC curves of XGBoost models predicting β-CTx, T-P1NP, and N-MID, respectively. (C), (F), and (I) Denote the P-R curves of XGBoost models predicting β-CTx, T-P1NP, and N-MID, respectively.

Supplemental Information 7 The RF model predicting BTMs (β-CTx/ T-P1NP/ N-MID).

(A), (D), and (G) Denote the confusion matrices of RF models predicting β-CTx, T-P1NP, and N-MID, respectively. (B), (E), and (H) Denote the ROC curves of RF models predicting β-CTx, T-P1NP, and N-MID, respectively. (C), (F), and (I) Denote the P-R curves of RF models predicting β-CTx, T-P1NP, and N-MID, respectively.

Supplemental Information 8 The LGBM model predicting BTMs (β-CTx/ T-P1NP/ N-MID).

(A), (D), and (G) Denote the confusion matrices of LGBM models predicting β-CTx, T-P1NP, and N-MID, respectively. (B), (E), and (H) Denote the ROC curves of LGBM models predicting β-CTx, T-P1NP, and N-MID, respectively. (C), (F), and (I) Denote the P-R curves of LGBM models predicting β-CTx, T-P1NP, and N-MID, respectively.

Supplemental Information 9 The initial variables of the model.

We thank all participants who agreed to participate in the study.

Additional Information and Declarations

Competing Interests

The authors declare that they have no competing interests.

Author Contributions

Shunshun Cao conceived and designed the experiments, analyzed the data, authored or reviewed drafts of the article, and approved the final draft.

Aolei Chen conceived and designed the experiments, performed the experiments, prepared figures and/or tables, authored or reviewed drafts of the article, and approved the final draft.

Botian Song conceived and designed the experiments, performed the experiments, prepared figures and/or tables, authored or reviewed drafts of the article, and approved the final draft.

Yangyang Hu conceived and designed the experiments, analyzed the data, authored or reviewed drafts of the article, and approved the final draft.

Human Ethics

The following information was supplied relating to ethical approvals (i.e., approving body and any reference numbers):

The Ethics Committee of the Second Affiliated Hospital of Wenzhou Medical University approved the study in its facilities (Ethics Application Ref. 2024-K-031-02).

Data Availability

The following information was supplied regarding data availability:

The original data and code are available in the Supplemental Files.

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
