# Peer review of "Exploring the effect of the triglyceride-glucose index on bone metabolism in prepubertal children, a retrospective study: insights from traditional methods and machine-learning-based bone remodeling prediction"

_PeerJ, doi:10.7717/peerj.19483_

## Round 0.1 · original submission · Major Revisions

Please revise your manuscript in line with the reviewer comments.

·

Basic reporting

What were the main inclusion and exclusion criteria in this article?
It seems to be better to add an inclusion and exclusion criteria diagram

Experimental design

The sample size of this study was 332 children. How was determined the sample size determined?
The numbers of the two study groups were partially the same. Is gender important in this article? If yes, how?

Validity of the findings

What was the association between age and BMI with the result of this study?
Is this sample size sufficient to generalize the results?
In the method section, it was mentioned that the age range of 7-11 years was chosen because this is a critical period of growth and development for children. Since this range was the same for boys and girls, the aim of this study was about prepubertal children. Please introduce this concern.

Additional comments

The style of writing was not justified. Please check the author’s guidelines. What was the association between age and BMI with the result of this study? Additionally, some of the Following references can be included in the introduction part for more readability:
https://doi.org/10.1016/j.ejim.2024.08.026
https://doi.org/10.1002/osp4.758
https://doi.org/10.1002/jcla.24995
https://doi.org/10.3390/nu15163507
https://doi.org/10.1111/jcmm.17646

·

Basic reporting

The authors seek to use the TyG index as a measure of insulin resistance to assess the potential of this index to inform about changes in bone health of obese prepubertal patients. To assess bone health they use bone turnover markers, relating their values to the TyG index and weight . They evaluate multiple machine learning algorithms to explore the relationship between TyG index and bone turnover markers. The link in the pediatric population between TyG index, insulin resistance and metabolic or cardiovascular disorders is not well documented (omitting key references such as Aslan Çin NN, Yardımcı H, et al. 2020). Nor do they review data that links insulin resistance to bone in the pediatric population (omitting references such as Bilinski). They state the concept but the references are not provided. They also fail to link bone turnover markers to a measure of bone fragility or any other comorbidity of obesity, only providing references that demonstrate BTM relationship to bone remodeling rate and metabolism but not associated or correlated with insulin resistance or metabolic disorders.

Experimental design

The experimental design is retrospective observational and is appropriate for an exploratory study. It represents primary research as there is limited data on the topic in the pediatric age group, though some evidence is present and omitted from the references provided by the authors (Bilinski WJ, et al, 2021)
However the research question involves prepubertal children which are only here defined by age. No physical exams are documented to insure the population is prepubertal which is critical since normal onset of puberty occurs as early as 8 in girls and 9 in boys.

Validity of the findings

Findings appear appropriate to the questions asked, though the analysis only leads to weak conclusions. More effort appears to be used to explore multiple machine learning algorithms than to establishing how sound the results are to the questions posed by the research. A less detailed explanation of each machine learning model with a better description of the most effective, or the summed conclusion of the models would enhance understanding of the paper.

Given the overall goal it does not seem necessary to include all the graphs for each model. Table 3 and the results of the SHAP analysis seem sufficient. Other data should be in supplemental materials.

Additional comments

Ln 58 Prevalence of obesity cited is for US but the sentence begins with identifying obesity as a global health risk

Have not included the reference by Rand, M et al 2023 that has developed sex and age-specific cut offs for BTM and related to future change in BMD

---

## Round 0.2 · accepted · Accept

Dear Dr. Hu,

Thank you for submitting the revised version of your manuscript. After a thorough review of the changes by the reviewers and me, I am pleased to inform you that all the reviewers' comments have been adequately addressed. Therefore, your manuscript is ready for publication in PeerJ.

Sincerely yours,
Stefano Menini